# Neurocognitive Suicide and Homicide Markers in Patients with Schizophrenia Spectrum Disorders: A Systematic Review

**DOI:** 10.3390/bs13060446

**Published:** 2023-05-26

**Authors:** Mario Tomé-Fernández, Marina Berbegal-Bernabeu, Miriam Sánchez-Sansegundo, Ana Zaragoza-Martí, María Rubio-Aparicio, Irene Portilla-Tamarit, Lorena Rumbo-Rodríguez, Jose Antonio Hurtado-Sánchez

**Affiliations:** 1Department of Health Psychology, Faculty of Health Science, University of Alicante, 03690 Alicante, Spain; mtf26@alu.ua.es (M.T.-F.); marina.berbegal@ua.es (M.B.-B.); maria.rubio@ua.es (M.R.-A.); irene.portilla@ua.es (I.P.-T.); ja.hurtado@ua.es (J.A.H.-S.); 2Department of Nursing, Faculty of Health Science, University of Alicante, 03690 Alicante, Spain; ana.zaragoza@ua.es (A.Z.-M.); lrr51@gcloud.ua.es (L.R.-R.)

**Keywords:** suicide, homicide, neurocognition, schizophrenia, neuropsychology, systematic review

## Abstract

Suicide and homicide are considered important problems in public health. This study aims to identify the cognitive performance of suicidal and homicidal behaviors in people with schizophrenia spectrum disorders, as well as examining whether there are shared neuropsychological mechanisms. A systematic review of the recent literature was carried out from September 2012 to June 2022 using the Medline (via PubMed), Scopus, Embase, and Cochrane databases. Among the 870 studies initially identified, 23 were finally selected (15 related to suicidal behaviors and 8 to homicidal behaviors). The results evidenced a relationship between impairment of cognitive performance and homicidal behavior; meanwhile, for suicidal behaviors, no consistent results were found. High neuropsychological performance seems to act as a protective factor against violent behavior in people with schizophrenia spectrum disorders, but not against suicidal behavior; indeed, it can even act as a risk factor for suicidal behavior. To date, there is insufficient evidence that shared neurocognitive mechanisms exist. However, processing speed and visual memory seem to be affected in the presence of both behaviors.

## 1. Introduction

Suicide and homicide are considered important problems in public health. Suicide is defined as a self-inflicted act that aims to cause death voluntarily [1], whereas homicide is defined as the act of killing a person without premeditation or another aggravating circumstance. According to the World Health Organization (WHO) [2], suicide has increased considerably in recent decades, reaching truly worrying levels. Approximately one million people die every year due to suicidal behavior (one person every forty seconds) and the number of attempts is estimated to be twenty times higher [3]. Death by suicide is the third leading cause of violent death among people aged from 15 to 44, behind only traffic accidents and homicides [4]. In the case of homicide, the total number of cases worldwide has also increased in recent decades, registering 464,000 homicides in 2017. Europe is the continent with the second lowest homicide rate, with 3 deaths per 100,000 inhabitants (22,009 total), 5% of the world total. In Spain, the average is less than 1% [5].

Biological, genetic, psychological, and environmental markers have been highlighted as factors related to suicidal and homicidal behaviors [6,7]. From a psychological point of view, the most important risk factor is the presence of psychiatric disorders [1,8], especially schizophrenia [9,10]. Schizophrenia is defined as a serious mental disorder characterized by an abnormal perception of reality, disorganized thought, maladaptive behaviors, and negative symptoms that affect the cognitive, emotional, and social domains [11].

The literature has found that suicide is the leading cause of premature death in patients with schizophrenia [12]. Between 4% and 13% of patients commit suicide, and approximately 60% have made at least one attempt [13]. In addition, between 6% and 11% of people with schizophrenia spectrum disorders exhibit homicidal behaviors [14]. A systematic review carried out by Hor and Taylor [15] concludes that the most significant risk factors related to suicide are being male, young, having a high level of education, and manifesting active symptoms of depression, hallucinations, and a good capacity for introspection (insight). For violent behavior, being male, the young/early onset of illness, having undergone numerous hospitalizations [16], past criminality [17], substance abuse such as alcoholism unemployment, psychotic symptoms [18], delusions and hallucinations, poor adherence to medication [19], and a lack of insight [20] are some of the most significant risk factors.

Recent research on cognition and violent behavior has reported a relationship between dysfunctional cognitive function and violent or homicidal behavior, but these variables have received less attention in the literature. Notably, it has been reported that individuals with lower intelligence, worse memory function and verbal learning, and, to a lesser extent, executive function, attention, and processing speed [21,22] exhibit more violent behaviors. Nevertheless, evidence about the role of cognitive functioning and suicidal behavior is contradictory. Thus, while a systematic review [23] showed that suicide attempters with schizophrenia outperformed non-attempters in executive functioning, attention, and verbal memory, other studies did not find any differences between individuals at high risk of suicide and homicide [24,25,26]. Moreover, some studies have shown that higher cognitive performance, and particularly executive functions, could imply a stronger ability to initiate suicidal behaviors [26].

More recently, it has been reported that suicide and homicide may share structural alterations in the brain that predispose individuals to violence and criminal and suicidal behavior [27,28]. The two areas involved in this relation that have received the most empirical support in neuroimaging and cognitive functioning studies are the prefrontal cortex, which is located in the anterior part of the frontal lobe, and the limbic system, which is located in the medial temporal lobe [29]. The first part is responsible for executive functions, which are cognitive processes that aim to consciously control and coordinate thoughts, emotions, and actions to achieve goals [30]. Likewise, the prefrontal cortex is divided into three areas: the dorsolateral, orbitofrontal, and anterior cingulate parts. The dorsolateral cortex has received attention in recent years, since it is responsible for processing sensory and emotional information, and organizing behavior through skills such as planning, cognitive flexibility, working memory, and inhibition [31]. Meanwhile, the limbic system is made up of structures that are responsible for emotional processing and regulation, the anticipation of future consequences, and reward and punishment mechanisms [32]. The literature suggests that the neuroanatomical and neuropsychological deficits described in schizophrenia manifest before positive and negative symptoms and remain stable over time [33]. These findings indicate that the global deficit in schizophrenia could be the result of multiple functional and structural alterations in the brain that not only influence the appearance of symptoms of psychosis, but also predispose patients to violence and suicide [34,35].

However, to date, research has not been able to clarify whether suicidal and homicidal behaviors in people with schizophrenia spectrum disorders arise due to shared or distinct neurocognitive mechanisms. Therefore, this study aims to identify the cognitive functions of suicidal and homicidal behaviors in people with schizophrenia spectrum disorders; additionally, it examines whether there are any shared neuropsychological mechanisms. The present systematic review of the recent literature was carried out to synthesize those articles that evaluated the association between neurocognitive markers and suicidal and homicidal behaviors.

(a)What is the relationship between cognitive functions and suicide and homicide behaviors in patients with schizophrenia spectrum disorders?(b)Does the performance of cognitive functions predispose to these behaviors?(c)Are there shared neurocognitive mechanisms?

## 2. Materials and Methods

This systematic review was conducted following the Preferred Reporting Items for Systematic Reviews and Meta-analyses (PRISMA) guidelines [36].

### 2.1. Eligibility Criteria

The inclusion criteria were as follows: articles that (1) were published in a peer-reviewed journal; (2) were published in English; (3) included people with schizophrenia, brief psychotic disorders, not otherwise specified (NOS) psychosis, or schizoaffective, schizophreniform, and delusional spectrum disorders diagnosed as per the DSM-IV, DSM-IV-SCID, or DSM-V criteria; (4) included at least one neuropsychological task; and (5) compared at least two groups of patients, one of which comprised patients with a history of suicide attempts (defined as any act carried out with a certain intent to die or to put one’s life in danger [37]), or suicidal ideation (defined as a range of contemplations, wishes, and preoccupations related to death and suicide [38]), or with a history of homicide (when there is a dead person and the cause of death can be attributed to another person [39]), or violence (violent acts committed against others which cause or are intended to cause physical harm to the victim [40]).

### 2.2. Information Sources

A systematic search was carried out using the Medline (Pubmed), Scopus, Embase, and Cochrane databases. Furthermore, to identify the potential additional studies not contained in the electronic databases, we performed a manual search of the bibliographical references of retrieved studies and previous systematic reviews [7,8,23].

### 2.3. Search Strategy and Selection Process

We performed a systematic literature search of all clinical trials, as well as cohort, case–control, and cross-sectional human studies published in English from 1 September 2012 to 1 June 2022. The search processes were undertaken blind by M.T.-F. and M.B.-B. in the Medline (via Pubmed), Scopus, Embase, and Cochrane databases. The MeSH terms “schizophrenia” and “psychotic disorders” were combined with the MeSH terms “violence”, “homicide”, “suicide”, and “attempted suicide” in all fields. Then, these were combined with the MeSH terms and TIAB terms of neurocognitive pathways: “cognition”, “neuropsychology”, “neuropsychological test”, “executive function”, “decision making”, “problem solving”, “prefrontal cortex”, “neuropsychological functions”, “executive functioning”, and “executive performance”. Table 1 and Table 2 show the full search strategy used in the review.

Each study identified through the literature search was independently evaluated by two researchers (M.T.-F. and M.B.-B.). In the identification phase, the total results were added to the Mendeley platform, and duplicates were removed. Afterwards, in the screening phase, the authors identified articles that met the inclusion criteria by validating their titles and abstracts. Subsequently, following a detailed reading of full-text articles, the authors determined the selections. In the inclusion phase, the chosen articles were identified and prepared for data extraction. The complete selection processes divided into suicide and homicide studies are reflected in Figure 1 and Figure 2.

### 2.4. Data Collection Process

All data were independently extracted by the authors (M.T.-F. and M.B.-B.) and the description and characteristics of each study were considered (authors, year of publication, country, objectives, average age, sample number, and type of study) (Table 3 and Table 4), and the design and results (diagnoses, cognitive domains and tasks, and the main results) were obtained (Table 5 and Table 6). The statistical measures that were taken into consideration were: *p*-value and effect size (eta partial square, Cohen’s d, and odd ratio).

### 2.5. Risk of Bias Assessment of Individual Studies

The quality of the included studies was assessed based on the Appraisal tool for Cross-Sectional Studies (AXIS) [41], which was developed for use in appraising observational cross-sectional studies. In addition, we used the JBI checklist for longitudinal cohort and case–control studies to assess their methodological quality and to determine the extent to which a study addressed the possibility of bias in its design, conduct, and analysis (Appendix A and Appendix B). It was carried out independently by two authors (M.T.-F. and M.B.-B.). In general, the quality of the cross-sectional and longitudinal studies included was quite good (Table A1, Table A2 and Table A3). Disagreements were resolved by a third author (M.S.-S.).

### 2.6. Synthesis Methods

The systematic literature review identified a total of 23 articles. The results were grouped based on whether the studies explored neurocognitive functioning in homicidal (8) or suicidal (15) individuals with schizophrenia spectrum disorders.

Regarding the suicide research, we added one study that was identified through the references of a previous systematic review. In addition, there was one study which appeared in the homicide search strategy that pertained to the suicide criteria. That is the reason why one is subtracted and added to the other group.

### 2.7. Certainty Assessment

The studies were assessed for their methodological rigor using criteria established by the Oxford Centre for Evidence-based Medicine levels of evidence for diagnostic studies. A summary of the methodology rigor is calculated based on the grade of level evidence for each study (from level 1 “systematic review, validating cohort studies” to level 5 “expert opinion”) [42].

## 3. Results

### 3.1. Selection of Studies: Suicide

A total of 599 studies were identified (598 through databases and 1 through another source). After eliminating duplicates, 391 studies remained. First, the titles and abstracts were screened, and 49 studies were excluded according to the different exclusion criteria. Second, 342 full-text articles were assessed for eligibility, and 328 of them were eliminated based on the different exclusion criteria. One study related to suicide was included in the systematic literature search for homicide; for this reason, the study was added to the total number of suicide articles. Finally, 15 articles related to suicide were included in this review with a total sample of 5118 patients (Figure 1).

### 3.2. Selection of Studies: Homicide

A total of 271 studies were identified (all obtained through databases). After eliminating duplicates, 157 studies were left. First, the titles and abstracts were screened and 43 were excluded according to the different exclusion criteria. Then, 114 full-text articles were assessed for eligibility; 105 of them were eliminated based on the different exclusion criteria. The final number of articles was 9; however, one of these was related to suicide, so it was included into the final list of suicide articles. Finally, 8 articles related to homicide were included in this review, with a total sample of 3412 patients (Figure 2).

### 3.3. Descriptive Data and Types of Studies

Table 3 and Table 4 show the characteristics of the articles included. Of the suicide studies, 63.9% (*n* = 3268) of participants were men and 30.1% (n = 1540) were women (N = 5118), while the sex of the participants was not specified in the remaining 6.1% (n = 310) of the sample. The mean age of the participants was 36.9 years old, and age was not specified in the remaining 6.6% (n = 1) of the sample. In the case of homicide studies, 65.6% (n = 2240) of the participants were men and 34.4% (n = 1172) were women (N = 3412). The mean age of the participants was 38.7 years old, and age was not specified in one article, which corresponds to 11.6% (n = 398) of the sample.

Regarding the country of origin, four studies were conducted in Spain [43,44,45,46] and six in China [47,48,49,50,51,52]. Two studies were conducted in the United States [53,54], two studies were carried out in the United Kingdom [55,56], two in Italy [57,58], two in Ireland [13,59], and two in Japan [60,61]. Finally, one study was conducted in India [62], one in Norway [63], and one in Turkey [64].

Appendix A and Appendix B (Table A1, Table A2 and Table A3) also identify the designs of the studies, showing that twenty of them were cross-sectional, while only four were longitudinal studies: 50% from suicide (n = 2) [44,45] and 50% from homicide (n = 2) [55,59].

### 3.4. Risk of Bias in Studies

Appendix A and Appendix B summarized the risk of bias for individual studies. Overall, the quality of studies was rated as medium-high in 15 of the 19 examined studies (78.9%), with only 4 studies rated as medium-low quality or doubtful (21.1%) [13,51,64]. A summary assessment was calculated based on the number of items evaluated as a *Yes* in the AXIS tool. Values above average were interpreted as medium-high quality and values below average were interpreted as medium-low quality. For longitudinal studies, the risk of bias was rated as low, with average values of 9 in items evaluated with the JBI tool.

### 3.5. Certainty of Evidence

The results of the methodological rigor by using the criteria set established by the Oxford Centre for Evidence-Based Medicine are presented in Figure 3. A limited number of highly methodologically rigorous studies were found, with only two homicide longitudinal studies [55,59] and two suicide longitudinal studies [44,45] classified within level 1b. No level 2 studies were found in either homicide or suicide. Most homicide and suicide studies were classified within the level of evidence 3b (moderate). Of the three levels, 5 homicide studies and 11 suicide studies were designed as non-randomized controlled cohort or local non-random samples without follow-up (Table 3). In addition, one homicide study [64] and two suicide studies [13,51] were found in the four levels as they were considered poor quality prognostic cohort studies.

### 3.6. Neurocognitive Functioning in Relation to Homicidal or Suicidal Behavior in Schizophrenia

The systematic literature review identified a total of 23 articles. Eight of these studies explored neurocognitive functioning in homicidal patients with schizophrenia, and fifteen analyzed neurocognitive performance in suicidal patients with schizophrenia. Table 6 and Table 7 show the diagnoses (100% of the included articles used DSM-IV or DSM-IV-SCID as diagnostic criteria), cognitive domains and task used in the investigations, and the main results of the included articles on suicide and homicide.

#### 3.6.1. Neurocognitive Functioning and Suicidal Behaviors in Patients with Schizophrenia

From the fifteen studies that explored suicidal behavior (Table 5), ten found differences between the groups and showed contradictory results. In four studies, the results showed that suicide attempters had more strongly affected cognitive functions than non-attempters, having impairments in global cognitive functioning (*p* < 0.01; F(1.134) = 7.10) (a GCF composite metric obtained using seven cognitive domains: verbal memory (RAVLT), visual memory (Rey complex figure), executive functioning (TMT-A/B), working memory (WAIS-III, backward digits scale), processing speed (WAIS-III digit symbol subtest), motor dexterity (GP), and attention (CPT)) and visual memory (*p* < 0.01 F(1.186) = 8.16) (Rey complex figure) [45], working memory (*p* < 0.05; OR = 0.04) (BACS) [52], processing speed (*p* = 0.04) (WAIS-III digit symbol subtest) [44], and decision-making (*p* < 0.01; η_P_^2^ = 0.19) (the Iowa Gambling Test) [43].

In the other six studies, suicide attempters with schizophrenia tended to outperform non-attempters with schizophrenia in premorbid IQ (*p* < 0.01; F = 7.34) (NART) [56], executive functioning (TMT-A *p* = 0.02; TMT-B *p* = 0.01) (TMT-A/B) [62], attention (*p* = 0.02; η_P_^2^ = 0.49) (RBANS, digit span and coding task) [51], planning (*p* = 0.000/Cohen’s d = 1.25; *p* = 0.006/Cohen’s d = 0.73) (Zoo map part 1; part 2) [46], IQ (*p* = 0.01; H = 6.18) (WAIS-III), episodic memory (*p* = 0.04; H = 0.92) (WMS-III), and working memory (*p* = 0.04; H = 1.02) (WMS-III letter–number frequency) [13] and verbal learning (*p* < 0.01; Cohen’s d = 0.49) (MCCB) [54]. Those who had a history of active suicide ideation also performed better than those who did not in the MCCB total score (*p* = 0.02; Cohen’s d = 0.363), verbal learning (*p* < 0.01; Cohen’s d = 0.482), working memory (*p* = 0.01; Cohen’s d = 0.397), and processing speed (*p* = 0.0; Cohen’s d= 0.334) (MCCB) [54]. However, the last five studies did not show statistical differences between the groups [47,48,49,50,52].

#### 3.6.2. Neurocognitive Functioning and Homicidal Behaviors in Patients with Schizophrenia

Of the eight studies that analyzed homicidal and violent behaviors (Table 6), six found statistical differences between the groups, showing that those with homicidal behaviors were more affected in cognitive functions than those without homicidal behaviors. This finding was particularly evident for verbal learning (*p* < 0.01; η_P_^2^ =0.08; Cohen’s d = 0.92) (MCCB) [59], fluid intelligence (*p* = 0.02; η_P_^2^ = 0.006) (WAIS-III Block Design subtest), and planification (*p* = 0.02; η_P_^2^ = 0.01) (NAB Mazes Test) [55], visual memory (*p* = 1.9 × 10^−5^; Cohen’s d = 0.34) (WAIS-III) [60], processing speed (*p* < 0.01; Cohen’s d = 0.41) (BACS symbol coding task) [58], IQ (*p* = 0.13; Cohen’s d = 0.52) (WASI), verbal learning (*p* = 0.03; Cohen’s d = 0.82) (HVLT-R) [63], and verbal memory (*p* = 0.01) (CVLT) [64].

Just one of the studies showed that homicidal patients outperformed non-homicidal patients in working memory (*p* = 0.04; Cohen’s d = 0.61) (BACS–Digit Sequencing Test) and executive function (*p* < 0.001; Cohen’s d = 1.00) (BACS total score) [61]. Finally, one study did not show statistical differences between the groups [57].

#### 3.6.3. Shared Neuropsychological Impairments in Suicidal and Homicidal Patients

The results showed that there are cognitive domains shared that significantly affect homicidal and suicidal behaviors. In Table 7, it can be seen how a worse performance in processing speed [44,58], and visual memory [45,60] could explain both violent behaviors (homicide and suicide).

This section may be divided by subheadings. It should provide a concise and precise description of the experimental results, their interpretation, as well as the experimental conclusions that can be drawn.

**Table 3 behavsci-13-00446-t003:** Description of the studies of suicide included in the review.

Ref.	Country	OCEBM *	Year	Objective	Average Age	Sample No. (N)	Type of Study
[48]	China	Moderate	2021	To investigate the lifetime suicide attempt rate, clinical characteristics, and cognitive function of Chinese patients with chronic schizophrenia who had attempted suicide.	46.4	N = 908 (742M/166F)	Cross-sectional study
[53]	USA	Moderate	2021	To examine the relationships between positive and negative symptoms, symptoms of depression, clinical insight, cognitive functioning, and suicidal ideation among a first-episode sample of participants with psychosis.	23.6	N = 404 (293M/111F)	Cross-sectional study
[46]	Spain	Moderate	2020	To examine the relationship between neurocognitive functioning and the history of suicidality in violent offenders with schizophrenia.	44.3	N= 61 (M)	Cross-sectional study
[49]	China	Moderate	2020	To examine whether there is an effect of DβH 5′-insertion/deletion (Ins/Del) polymorphism on cognitive performance in suicide attempters with chronic schizophrenia.	47.5	N = 731 (615M/116F)	Cross-sectional study
[50]	China	Moderate	2019	To investigate the prevalence of suicidal ideation in patients with schizophrenia, and to identify which clinical symptoms and biochemical parameters were most strongly associated with suicidal ideation.	36.2	N = 174 (82M/92F)	Cross-sectional study
[51]	China	Low	2018	To examine the prevalence of suicide attempts and the association of this prevalence with demographic and clinical variables and cognitive function in Chinese first-episode, drug-naive (FEDN) schizophrenia patients.	27.5	N = 737 (352M/385F)	Cross-sectional, case control design
[45]	Spain	Very high	2018	To explore predictors of suicidal behavior, adjusting the analyses for a set of sociodemographic, clinical, and neurocognitive variables. Additionally, to examine potential long-term differences in clinical measures and neuro- cognitive functioning between patients who undertook suicidal acts and those who did not over the follow-up period.	29.9	N = 517 (297M/220F)	Longitudinal study
[54]	USA	Moderate	2018	To assess for cognitive ability, cognitive insight, and a history of suicidal ideation and behavior.	50.6	N = 162 (86M/76F)	Cross-sectional study
[43]	Spain	Moderate	2017	To explore the differences in executive functioning between suicide attempters and non-attempters in dual schizophrenia patients, and the possible premorbid and clinical-related factors.	36.1	N = 50 (M)	Cross-sectional study
[62]	India	Moderate	2016	To investigate the triangular relationship between suicide intent, insight, and cognitive competence in schizophrenia.	33.5	N = 175 (107M/68F)	Cross-sectional study
[44]	Spain	Very high	2015	To examine the premorbid, demographic, clinical, insight, and neurocognitive characteristics that are potentially related to suicide risk before the first presentation to psychiatric services and over the follow-up period.	28.9	N = 397 (226M/171F)	Longitudinal study
[52]	China	Moderate	2014	To test the hypothesis that higher cognitive function is associated with an increase in suicide attempts in a population of Han Chinese patients suffering from schizophrenia.	51.6	N = 316 (236M/80F)	Cross-sectional pilot study
[47]	China	Moderate	2014	To examine the prevalence of suicidal ideation and its relationship with clinical, neurocognitive, and psychological factors in first-episode psychosis patients.	20.5	N = 89 (43M/46F)	Cross-sectional study
[56]	UK	Moderate	2013	To investigate the demographic, clinical, and neuropsychological aspects of self-harm in schizophrenia, and to identify which of these aspects are independently predictive of and therefore the most relevant to clinical intervention.	39.7	N = 87 (78M/9F)	Cross-sectional study (prospective)
[13]	Ireland	Low	2012	To investigate whether the relationship between suicidality and neurocognition varied according to differences in suicidal ideation and behavior.	-	N = 310	Cross-sectional study

* OCEBM: Oxford Centre for Evidence-Based Medicine.

**Table 4 behavsci-13-00446-t004:** Description of the studies of homicide included in the review.

Ref.	Country	OCEBM *	Year	Objective	Average Age	Sample No. (N)	Type of Study
[60]	Japan	Moderate	2022	To provide a resource for risk assessment and intervention studies by conducting multifaceted well-established assessments.	35.3	N = 1620 (834M/786F)	Cross-sectional study
[58]	Italy	Moderate	2021	To analyze the differential predictive potential of neurocognition and social cognition to identify patients with schizophrenia spectrum disorders with and without a history of severe violence.	Age 18–29 = 102; Age 30–41 = 153; Age 42–53 = 85; Age 54–65 = 58	N = 398 (336M/62F)	Cross-sectional study
[55]	UK	Very high	2020	To investigate the association between neuropsychological test performance and a sensitive marker of violent behavior.	26.8	N = 891 (688M/203F)	Longitudinal study
[63]	Norway	Moderate	2018	To investigate global and specific cognition among homicide offenders with schizophrenia (HOS).	36.3	N = 205 (126M/79F)	Cross-sectional study
[57]	Italy	Moderate	2017	To investigate the relationship between clinical and neuropsychological factors and violence risk in patients with schizophrenia, taking into account current psychopathology and lifetime alcohol use.	47.9	N = 87 (78M/9F)	Cross-sectional study (prospective)
[64]	Turkey	Low	2016	To investigate factors associated with violent behavior in schizophrenia and to clarify the relationship between violent behavior, insight, and cognitive functions.	42.2	N = 68 (40M/28F)	Cross-sectional study
[61]	Japan	Moderate	2015	To examine the backgrounds and neurocognitive functions of violent and nonviolent patients with schizophrenia to identify factors associated with serious violence.	42.2	N = 54 (M)	Cross-sectional study
[59]	Ireland	Very high	2015	To examine whether neurocognition and social cognition predicts inpatient violence amongst patients with schizophrenia and schizoaffective disorder over a 12-month period.	40	N = 89 (84M/5F)	Longitudinal study

* OCEBM: Oxford Centre for Evidence-Based Medicine.

**Table 5 behavsci-13-00446-t005:** Diagnoses, cognitive domains/tasks, and main results of the included articles on suicide.

Ref.	Year	Sample Groups	Diagnoses	Cognitive Domains/Task	Main Results
[48]	2021	908 individuals. Suicide: n = 97; non-suicide: n = 811	Schizophrenia (DSM-IV).	RBANS (repeatable battery for the assessment of neuropsychological status): immediate memory, visuospatial skills, language, attention, and delayed memory.	There were no significant statistical differences between the suicidal and non-suicidal groups.
[53]	2021	404 individuals. Suicide attempters: n = 106; non-suicide attempters: n = 298	Schizophrenia/schizoaffective disorder/schizophreniform disorder/brief psychotic disorder/delusional disorder or psychotic disorder. (DSM-IV).	BACS (brief assessment of cognition in schizophrenia): verbal memory, working memory, motor speed, verbal fluency, attention and speed of information processing, and executive function.	Clinical insight (*p* = 0.031; OR = 0.73) and working memory (*p* = 0.041; OR = 0.04) were associated with increased odds of suicide ideation after baseline.
[46]	2020	Suicide attempters: n = 26; suicide non-attempters: n = 35	Schizophrenia, schizoaffective disorder, or delusional disorder (DSM-IV-TR).	Premorbid IQ; national adult reading test (NART); current IQ (namely, verbal (vocabulary KBIT); nonverbal (matrix KBIT); attentional control (d2); episodic, verbal, and working memory (subsets of the Wechsler Memory Scale, third edition (WMS-III); subscale of letters and numbers (WAIS-III)); executive functioning (the Wisconsin Card-Sorting task (WCST); the computerized Tower of London test; the Zoo Map subset (BADS); the trail making test; the Stroop color–word task); planning abilities (the computerized Tower of London test; the Zoo Map subset (BADS); the Stroop color–word task); verbal fluency (controlled oral word fluency task (FAS)).	There were no significant statistical differences between the two groups in neurocognitive functioning. However, after controlling for the effects of demographic and clinical variables, suicide attempters performed better than suicide non-attempters in two planning-related tasks (in both tasks: *p* < 0.001): the Zoo Map Part 1 (*p* = 0.000; Cohen’s d = 1.25), Part 2 (*p* = 0.006; Cohen’s d = 0.73), the Tower of London extra moves (*p* = 0.000; Cohen’s d = 2.73), and the Tower of London time: seconds (*p* = 0.002).
[49]	2020	731 patients. Attempters: n = 114; non-attempters: n = 617	Schizophrenia (DSM-IV).	RBANS (repeatable battery for the assessment of neuropsychological status): immediate memory, list learning, story memory, attention, digit span, coding, language, picture fluency, visuospatial, figure copy, line orientation, delayed memory, list recall, story recall, figure recall, and list recognition	There were no significant statistical differences between attempters and non-attempters.
[50]	2019	Suicidal ideation: n = 26; no suicidal ideation: n = 148	Schizophrenia disorder (DSM-IV).	Total score of RBANS (repeatable battery for the assessment of neuropsychological status): immediate memory (list learning and story memory tasks); visuospatial/constructional (figure copy and line orientation tasks); language (picture naming and semantic fluency tasks); attention (digit span and coding tasks); delayed memory (list recall; story recall; figure recall; list recognition tasks).	No significant statistical differences were found between those with suicidal ideation and those with no suicidal ideation in terms of performance on the RBANS test (total score and cognitive domains).
[51]	2018	Schizophrenia: n = 123; suicide attempters: n = 28; suicide non-attempters: n = 95; healthy controls: n = 151	Schizophrenia (DSM-IV—SCID).	Total score of RBANS (repeatable battery for the assessment of neuropsychological status): immediate memory (list learning and story memory tasks); attention (digits span and coding tasks); language (picture naming and semantic fluency tasks); visuospatial/constructional (figure copy and line orientation tasks); delayed memory (list recall, story recall, figure recall, and list recognition tasks).	Both groups with schizophrenia showed significantly lower cognitive scores on RBANS total, immediate memory, attention, delayed memory (all *p* < 0.001) and language (*p* = 0.002), than healthy controls. However, when suicide attempters were compared with non-attempters within the schizophrenia group, attempters performed better only on the attention domain (*p* = 0.025; η_P_^2^ = 0.49).
[45]	2018	Non-suicidal behavior: n = 466; suicidal behavior: n = 51	Schizophrenia, brief psychotic disorder, not otherwise specified (NOS) psychosis, schizophreniform disorder, schizoaffective disorder, delusional disorder (structured clinical interview for DSM-IV—SCID).	Global cognitive functioning (GCF); verbal memory (Rey auditory verbal learning test (RAVLT)); visual memory (Rey complex figure (RCF); delayed reproduction); executive functioning (trail making test (TMT)); working memory (WAIS-III backward digits subset); processing speed (WAIS-III digit symbol subtest); motor dexterity (grooved pegboard handedness (GP)); attention (continuous performance test (CPT)); premorbid IQ (WAIS-III vocabulary subtest).	Patients with suicidal behaviors presented worse scores in visual memory (*p* < 0.01; F(1.186) = 8.16) and global cognitive functioning (*p* < 0.01; F(1.134) = 7.10). In addition, global cognitive functioning (GCF) was the most important predictor of lifetime suicidality.
[54]	2018	No actual attempt: n = 95; attempt: n = 66	Schizophrenia/schizoaffective disorder (DSM-IV)	Total score of consensus cognitive battery (MCCB); verbal learning (Hopkins Verbal Learning Test); speed of processing (Trail Making Test A; symbol coding (Brief Assessment of Cognition in Schizophrenia); animal naming (category fluency)); working memory (letter–number span (WMS)); reasoning and problem solving (mazes (neuropsychological assessment battery)).	Patients with active suicidal ideation presented a greater MBCC total score (*p* = 0.025; Cohen’s d = −0.363), verbal learning (*p* = 0.003; Cohen’s d = −0.482), speed of processing (*p* = 0.038; Cohen’s d = 0.334), and working memory scores (*p* = 0.013; Cohen’s d = −0.397) than patients with non-active suicidal ideation. Additionally, patients with suicidal attempts performed better in verbal learning (*p* = 0.002; Cohen’s d = −0.49) than those without suicide attempts.
[43]	2017	Non-attempters: n = 26; Suicide attempters: n = 24	Dual schizophrenia/schizoaffective disorder (DSM-IV-TR).	Total score of premorbid IQ; vocabulary (WAIS-III); block design (WAIS-III); total score of executive functioning; working memory (backward digits (WAIS-III)); cognitive flexibility (trail making test (TMT) B); planning abilities (Tower of Hanoi); abstract reasoning/problem solving (WCST) and decision-making (Iowa gambling task).	Suicide attempters presented lower composite summary scores in executive function (*p* < 0.05; η_P_^2^ = 0.10), problem solving skills (*p* < 0.01; η_P_^2^ = 0.14), and decision-making (*p* < 0.01; η_P_^2^ = 0.19) compared to non-attempters. However, after controlling for the effects of alcohol dependence, only decision-making showed significant differences.
[62]	2016	Never attempted: n = 136 ever attempted: n = 39	Schizophrenia/schizoaffective disorder (DSM-IV)	Executive function (Trail Making Test (TMT) A and B)	The attempters scored significantly better in executive function on both TMT A (*p* = 0.026) and TMT B (*p* = 0.012) than those who had never attempted.
[44]	2015	Suicide attempters: n = 60; suicide non-attempters: n = 337	Schizophrenia, schizophreniform disorder, schizoaffective disorder, brief psychotic disorder, psychosis NOS, and delusional disorder. (DSM-IV).	Premorbid IQ (WAIS-III vocabulary); information processing speed (WAIS-III digit symbol); motor dexterity (grooved pegboard dominant hand); working memory (WAIS-III digits backward digits scale); verbal memory (RAVLT list delayed recall); visual memory (Rey figure delayed recall); attention (CPT); executive function (TMT B–A).	Processing speed was significantly (*p* = 0.046) more impaired in attempters. No significant differences were found in the other domains.
[52]	2014	Attempted suicide: n = 25 Non-attempters: n = 291	Schizophrenia disorder (structured clinical interview for DSM-IV—SCID).	Total score of RBANS (repeatable battery for the assessment of neuropsychological status): immediate memory (list learning and story memory tasks); visuospatial/constructional (figure copy and line orientation tasks); language (picture naming and semantic fluency tasks); attention (digit span and coding tasks); delayed memory (list recall; story recall; figure recall; list recognition tasks).	There were no significant statistical differences between the suicide attempters and non-attempters.
[47]	2014	Suicidal ideation: n = 37 No suicidal ideation: n = 52	Schizophrenia, schizophreniform disorder, delusional disorder, brief psychotic disorder, or psychosis not otherwise specified (DSM-IV).	Cognitive inflexibility (modified Wisconsin Card-Sorting test (MWCS)) and dyscontrol of executive inhibition (Hayline sentence completion test (HSCT) Part B).	There were no significant statistical differences between the suicidal and non-suicidal groups.
[56]	2013	87 patients. Self-harm: n = 59; no self-harm: n = 28	Schizophrenia (DSM-IV).	Nart (premorbid iq), trail making test (frontal executive function), computerized auditory continuous performance test (sustained attention and vigilance), computerized visual go/no-go reaction time task (cognitive–motor impulsivity) (240 go stimuli and 60 no-go stimuli).	Those with past self-harm, compared to those without, were significantly more likely to report impulsivity (*p* < 0.01; F = 7.97) and had higher premorbid IQs (*p* < 0.01; F = 7.34).
[13]	2012	Non-attempters with no ideation: n = 172; history of ideation without having made a suicide attempt: n = 63; a single attempt: n = 48; multiple attempts: n = 27	Schizophrenia/schizoaffective disorder (Structured Clinical Interview for DSM-IV- SCID).	Cognition (full scale IQ (WAIS-III); verbal IQ (WAIS-III); performance IQ (WAIS-III)); memory (logical memory 1 (WMS-III); logical memory 2 (WMS-III); PAL stages (CANTAB); PAL total errors (CANTAB)); working memory (letter–number (WMS-III); SWM errors (CANTAB)); attention (CANTAB IDED; CANTAB IDED-ED).	The “single attempters” group outperformed those in the “No Ideation, No Attempts” group in terms of current full-scale IQ (*p* = 0.02; H = 8.33) and verbal IQ (*p* < 0.05; H = 7.75). The “ideation only” group out-performed the “no ideation, no attempts” group in episodic memory (*p* < 0.03; H1.381). After regrouping, the “ideation only + single attempters” outperformed the “no ideation, no attempters” group in full-scale IQ (*p* = 0.01; H = 6.18), working memory (*p* = 0.04; H1.02), and episodic memory (*p* = 0.04; H = 0.92).

**Table 6 behavsci-13-00446-t006:** Diagnoses, cognitive domains/tasks, and main results of the included articles on homicide.

Ref.	Year	Sample Groups	Diagnoses	Cognitive Domains/Task	Main Results
[60]	2022	1620 individuals. Healthy subjects (HS): n = 1265; 355 patients with schizophrenia: history of violence (V-SZ): n = 112; without a history of violence (NV-SZ): n = 243	Schizophrenia. Structured Clinical Interview for DSM-IV Axis I Disorders (SCID-I).	Wechsler Adult Intelligence Scale (WAIS)-III, Wechsler Memory Scale–Revised (WMS-R), Wisconsin Card-Sorting Test (WCST), Verbal Fluency Test (VFT), Rey Auditory Verbal Learning Test (AVLT), and Continuous Performance Test–Identical Pairs version (CPT-IP).	There were significant differences between “Violent” and “Non-violent” in visual memory function (*p* = 1.9 × 10^−5^, Cohen’s d = 0.34), being lower in the V-SZ group.
[58]	2021	398 patients. Forensic patients: n = 221; non-forensic patients: N = 177	Schizophrenia/schizoaffective disorder. (DSM-V).	The Brief Assessment of Cognition in Schizophrenia (BACS). Verbal (list learning) and Working memory (Digit Sequencing Task), Motor speed (Token Motor Task), Verbal fluency (semantic and letter fluency), Attention and speed information processing (Symbol Coding Task) and Executive functions (Tower of London).	There were significant differences between “Forensic Group” and “Control group” in processing speed (BACS Symbol Coding Task: *p* < 0.01; η_P_^2^ = 0.49) with larger impairments in the Forensic group.
[55]	2020	891 patients. Violent: n = 183; non-violent: n = 708	Schizophrenia/schizoaffective disorder/psychotic disorder/others. (DSM-IV-TR).	Continuous Performance Test-HQ (CPT-HQ) (inhibition); Response Shifting Task (RST) (cognitive flexibility); Wechsler Adult Intelligence Scale, third edition (WAIS-III); Block Design subtest (fluid intelligence); (IV) Neuropsychological Assessment Battery (NAB): Mazes Test (planning); (v) Degraded Facial Affect Recognition Task (DFAR) (affective ToM); and (vi) Hinting Task (cognitive ToM).	Violent patients performed significantly worse than non-violent patients in fluid intelligence (*p* = 0.02; η_P_^2^ = 0.006), planning (*p* = 0.02; η_P_^2^ = 0.01), and theory of mind (cognitive part) (*p* < 0.01; η_P_^2^ = 0.01).
[63]	2018	Homicide offenders (HOS): n = 26; no history of violence (non-HOS): n = 28	Schizophrenia/schizoaffective disorder (ICD-10).	Global cognition (Vocabulary (WASI); Matrix Reasoning (WASI); MCCB comp); MCCB total score; Speed of processing (Trail Making Test (TMT)); Attention/vigilance (Continuous Performance Test–Identical Pairs); Working memory (Spatial Span and Letter–Number Span (WMS-III)); Verbal learning (Hopkins Verbal Learning Test–Revised (HVLT-R)); Visual learning (Brief Visuospatial Memory Test–Revised (BVMT-R)); Reasoning/problem solving Neuropsychological Assessment Battery (NAB): Mazes; Social cognition (Mayer–Salovey–Caruso Emotional Intelligence Test (MSCEIT): Managing Emotions); Color naming, Word Reading, Inhibition, inhibition/switching (Color–Word Interference Test (CWIT)).	The effects sizes for IQ were medium (Cohen’s d = 0.52), but the difference was no longer statistically significant (*p* = 0.13). For verbal learning, the group difference between HOS and non-HOS remained statistically significant (*p* = 0.03) and the effect size was large (Cohen’s d = 0.82).
[57]	2017	87 patients. Violent (vSZ): n = 50; non-violent (nvSZ): n = 37	Schizophrenia (DSM-IV).	Brief Assessment of Cognition in Schizophrenia (BACS): verbal memory, working memory, motor speed, verbal fluency, attention, and speed of information processing and planning. Wisconsin Card-Sorting test (WCST) (executive function: flexibility and inhibition). Iowa Gambling Test (IGT) (decision making).	The vSZ subjects had significantly higher motor speed scores (*p* = 0.007; η_P_^2^ = 0.03), lower executive function: flexibility (*p* = 0.023; η_P_^2^ = 0.03), and inhibition (*p* = 0.025; η_P_^2^ = 0.03) compared to the nvSZ group. However, entering the BPRS negative score and a lifetime problematic use of alcohol into the ANCOVA model, no significant differences between groups were found (*p* = 0.130, *p* = 0.122, and *p* = 0.114, respectively).
[64]	2016	68 individuals. Violent: n = 30; non-violent: n = 38	Schizophrenia. (Structured Clinical interview for DSM-IV axis I disorders (SCID-I)).	The California Verbal Learning Test (CVLT), Trail-Making Test (TMT), Wisconsin Card-Sorting Test (WCST) and Stroop test.	The non-violent group performed significantly better than violent group in verbal memory (CVLT long-delayed response: *p* < 0.05).
[61]	2015	Violent: n =30; non-violent: n = 24	Schizophrenia (DSM-IV-TR).	The Brief Assessment of Cognition in Schizophrenia (BACS): verbal memory (digit sequencing test); working memory (digit sequencing test); motor speed (token motor test); verbal fluency (symbol coding test); attention; executive functioning (BACS total score)).	The violent group performed significantly better than the control group on working memory (*p* = 0.047; Cohen’s d = 0.61) and executive function (*p* < 0.001; Cohen’s d = 1.00).
[59]	2015	89 patients. Violent: n = 10; non-violent: n = 79	Schizophrenia/schizoaffective disorder. Structured clinical interview for DSM-IV-TR.	The MATRICS Consensus Cognitive Battery (MCCB), which covers: processing speed, attention/vigilance, working memory, verbal learning, visual learning, and reasoning and problem solving; Test of Premorbid Functioning TOPF-UK.	The violent and non-violent groups differed only in the verbal learning domain (*p* = 0.007; η_P_^2^ = 0.08; Cohen’s d = 0.92).

**Table 7 behavsci-13-00446-t007:** Shared neuropsychological impairments in suicidal and homicidal patients.

Cognitive Domain	Homicidal Group	Suicidal Group
Processing speed	[58]	[44]
	BACS—symbol coding task	WAIS-III—digit symbol subtest
Visual Memory	[60]	[45]
	WAIS-III	Rey complex figure

## 4. Discussion

The present systematic review provides the most recent evidence on cognitive functioning in people with schizophrenia spectrum disorders who have committed suicidal and homicidal acts. Regarding the published studies pertaining to suicidal behaviors, the results were contradictory. Four of them [43,44,45,53] conclude that cognitive performance impairments predispose patients to suicidal ideation and suicide attempts. Among the significant cognitive domains that emerge in this research, three of them (working memory, global cognitive functioning (GCF), and decision making) are related to executive functions. These cognitive deficits, especially those related to working memory, have an impact on executive control, decision making, and higher reasoning skills, which are protective factors against suicidal ideation and attempts [53,65]. Additionally, recent studies on brain dysfunctions have reported reduced neural activity in suicide attempters who were exposed to fMRI (functional magnetic resonance imaging) [66,67].

Despite the heterogeneity shown by the selected articles, one trend that emerges shows cognitive performance to be a significant risk factor for suicidal behavior in patients with mental illness [13,46,51,54,56,62]. Previous studies concur that a higher educational level [68], higher IQs [69], better executive performance [24], and greater levels of insight [70] predispose patients to engage in suicidal behaviors. This fact could be explained by a patient’s greater understanding of the illness phase, since this could increase their awareness of their limitations and personal decline; additionally, it may raise their likelihood of suffering from positive symptoms such as depression [56]. In addition, there is a consensus that a better performance on cognitive functions, especially in executive domains such as planning, predispose patients to be better able to formulate plans and initiate goal-directed behavior to suicidal acts [71]. Neuroimaging studies also reaffirm this fact; they show greater activation in the prefrontal zone, an area responsible for the organization, planning, and initiation of behavior [29].

Moreover, in contrast to the results for suicide, homicide studies showed stronger findings. Six of the eight selected articles evidenced a relationship between a cognitive performance impairment and homicidal behavior [55,58,59,60,63,64]. These findings are supported by a recent meta-analysis across 43 studies [72], which concludes that violent behavior was related to structural and functional deficits in the prefrontal cortex. Alterations in the right orbitofrontal and cingulate anterior areas are associated with a lack of emotional processing, difficulties in social behavior, and poor decision making [73,74]. Deficits in the left dorsolateral cortex revealed a lower performance in attention, flexibility, planning, and impulse control [75,76,77].

Some studies reported that detrimental function could be due to reduced gray [78] and white matter volume [79] in the prefrontal cortex, temporal lobe, and superior temporal gyrus [80], as well as reduced blood oxygenation in the amygdala [81] and significantly lower activation in the frontal basal cortex area [82]. Furthermore, another imaging study examined processing speed impairments and found that this deficit was associated with damage in the cerebellar–thalamo–cortical circuit [83].

In addition, the systematic review showed that the shared mechanisms between both behaviors are not conclusive. Although there are two cognitive domains that appear as risk factors in both groups (processing speed and visual memory), there is another function that acts as a risk and protective factor depending on the behavior (working memory). Future research should compare the cognitive performance of the two groups with the same evaluation protocol, as it would be easier to establish relationships in this way.

There are several limitations of the current study that must be highlighted. First, the populations studied were highly heterogeneous in terms of their sociodemographic and clinical characteristics, especially in the primary diagnosis [44,45,46,47,53] and gender, since some studies only included male patients [43,46,61]. Additionally, there were missing sociodemographic data [13] and some studies included small sample sizes. Second, not only the gender or age of the sampled participants, but also the presence of polysubstance abuse (alcoholism, etc.), the duration of the illness, medication, psychotic symptoms, and hallucinations and other comorbidities (antisocial personality disorder) were variables not considered in some of the studies, despite these factors exerting a significant influence on the results. For example, Bulgari et al. [57] and Adan et al. [43] found significant results between the groups, but after controlling for these variables, the evidence was no longer significant. Third, the definitions of suicidal and homicidal behaviors were also heterogeneous, as well as the neuropsychological assessments used to evaluate the cognitive functions. For example, there were studies which used the WCST to measure abstract reasoning/problem solving [43], while others used it to evaluate executive function [46]. Moreover, some authors used a single neuropsychological test whereas others preferred a specific large battery. Fourth, the methodologies of the studies reviewed also varied (i.e., the tools used to assess suicidal and homicidal behaviors). Finally, this systematic review was not registered as a protocol. It will be considered for future research.

## 5. Conclusions

This article has brought to light the recent evidence on the relationship between cognitive functioning and suicidal and homicidal behaviors in people with schizophrenia spectrum disorders. These results could be used to give priority not only to psychological and pharmacological factors, but also to the prevention and treatment of neuropsychological domains. Our results suggest that cognitive function is affected in violent or homicidal patients with schizophrenia, showing a detrimental effect in cognitive functions such as learning, fluid intelligence, planification, visual memory, and processing speed. In particular, the suicide studies showed contradictory results. Therefore, no strong conclusions can be established regarding the influence of brain alterations or neuromarkers in suicide behaviors, and deeper investigations are required. When looking for shared pathways between suicidal and homicidal patients with schizophrenia spectrum disorders, there were statistically significant results with processing speed and visual memory. Nevertheless, future research that directly compares the neurocognitive markers of suicidal and homicidal risk are needed. Finally, the main contribution of this review is that it tries to understand if there are shared cognitive mechanisms which can help to identify and predict risk areas in crime intervention and treatment programs. Health institutions should prioritize not only psychological and pharmacological treatments, but also neuropsychological training.

## Figures and Tables

**Figure 1 behavsci-13-00446-f001:**
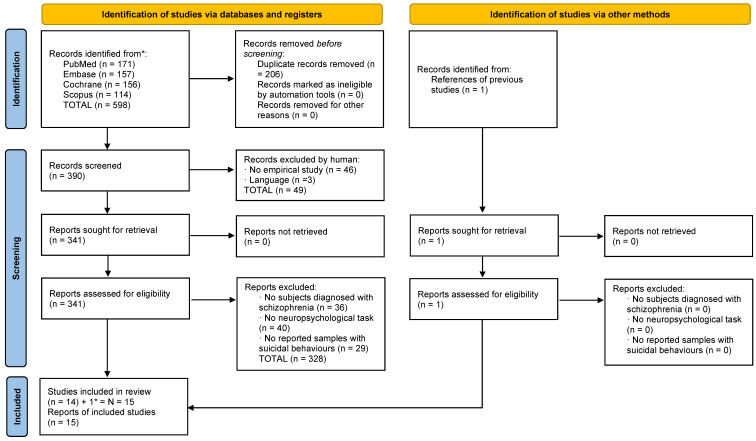
Flowchart of homicide studies. * it was added one study which appeared in the homicide search strategy but pertained to the suicide criteria.

**Figure 2 behavsci-13-00446-f002:**
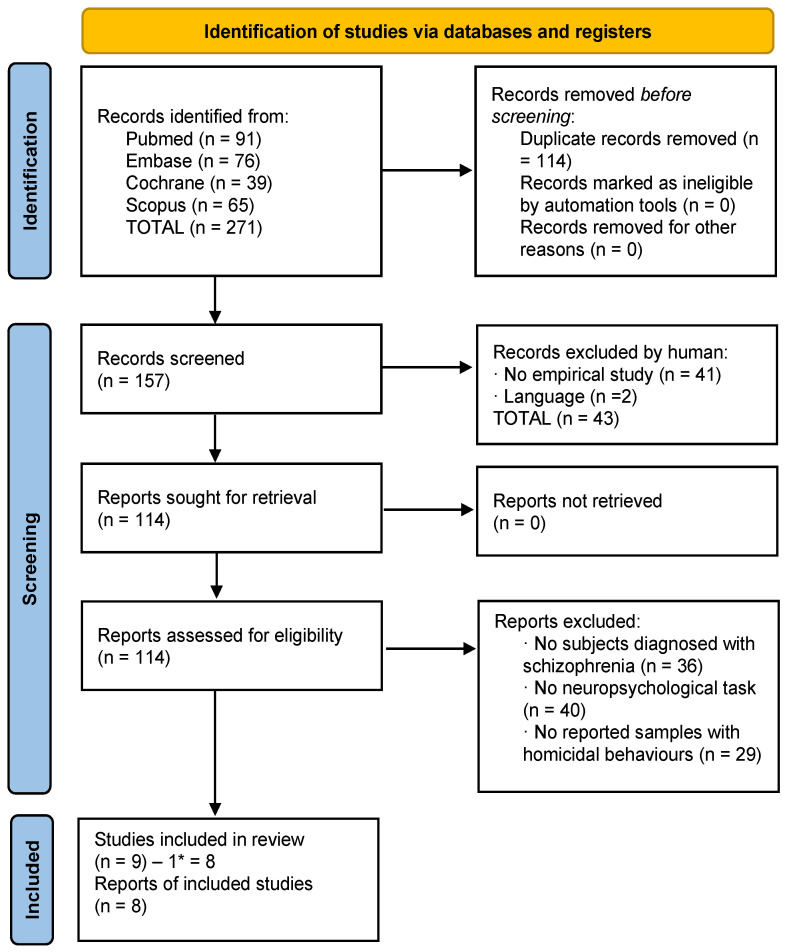
Flowchart of homicide studies. * One study was subtracted which pertained to the suicide criteria.

**Figure 3 behavsci-13-00446-f003:**
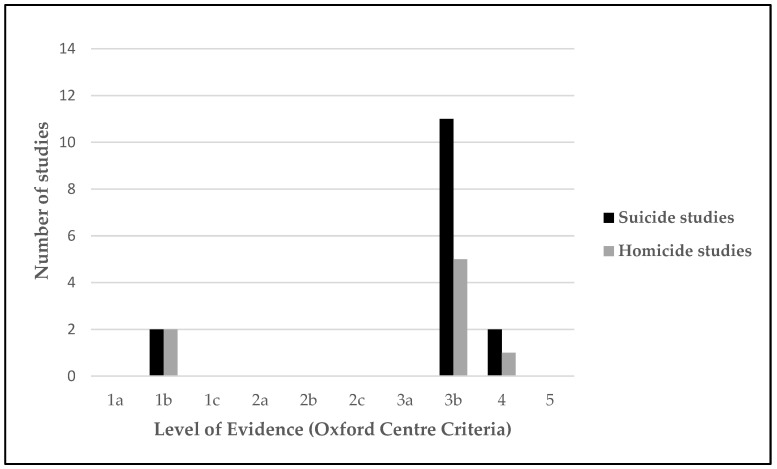
Level of evidence (Oxford Centre Criteria).

**Table 1 behavsci-13-00446-t001:** Search strategy used for the review.

Search Strategy	
1	“schizophrenia” OR “psychotic disorder”
2	“suicide” OR “attempted suicide”
3	“cognition” OR “neuropsychology” OR “neuropsychological test” OR “executive function” OR “decision making” OR “problem solving” OR “prefrontal cortex” OR “neuropsychological functions” OR “executive functioning” OR “executive performance”
4	#1 AND #2 AND #3

All terms set in quotations are MeSH terms.

**Table 2 behavsci-13-00446-t002:** Search strategy used for the review.

Search Strategy	
1	“schizophrenia” OR “psychotic disorder”
2	“homicide” OR “violence”
3	“cognition” OR “neuropsychology” OR “neuropsychological test” OR “executive function” OR “decision making” OR “problem solving” OR “prefrontal cortex” OR “neuropsychological functions” OR “executive functioning” OR “executive performance”
4	#1 AND #2 AND #3

All terms set in quotations are MeSH terms.

## Data Availability

Not available.

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
