# Peer review of "Neurocognitive Suicide and Homicide Markers in Patients with Schizophrenia Spectrum Disorders: A Systematic Review"

_behavsci, 2023, doi:10.3390/bs13060446_

Round 1
Reviewer 1 Report
The present study related to “Neurocognitive Suicide and Homicide Markers in Patients with Schizophrenia” is well written and it is full of information. However, majors edits must be made before being considered for publication in Behavioral Sciences.
Edits and suggestions :
1. Reference [33] is related to cognitive impairment in euthymic bipolar disorders not schizophrenia.
2. In introduction part, authors could add some recent references (2020-2022)
3. In 2.2 : add in the text Table 2 “Table 1 and table 2 show”.
4. In 2.4 : where are tables 1.2 and 2.1 mentioned in the text?
5. In 2.4: “Two authors (X.X. and X.X.)”….. “third author (X.X.)” please add the name of the authors.
6. In 2.5 : “two independent authors of the review (X.X. and X.X.)” please add the name of the authors.
7. Please insert Tables 4, 4.1, 5, 5.1, 6 and 7 in the text.
8. In 2.6: insert Table 3.1 next to “continued axis tool”
9. Please put the tables 3, 3.1, 4, 4.1, 5, 5.1, 7 in other format, we don’t see them all !
10. It would be better to insert the tables 3, 3.1, 4 and 4.1 in the results part
11. In tables 3, 3.1, 4, 4.1, 5, 5.1, 6 and 7, it would be better to insert the reference number instead of the author's name.
12. In 3.4.3. part, where is the Table 3.2 mentioned in the text ?
13. In discussion part, please add some recent references in studies related to fMRI
14. Finally the most important point : there are several important limitations of the study (diagnosis, gender, sociodemographic and clinical characterics, methodologies of the studies, definition of suicidal and homicidal behaviors, neuropsychological assessments used etc.) and no strong conclusion can be established, so what is the interest of this study and the originality?
Author Response
Thank you for giving us the opportunity to submit a revised draft of our manuscript. We appreciate the time and effort that you have dedicated to providing your valuable feedback on our manuscript. Here is a point-by-point response to the reviewers’ comments and concerns.
Point 1. Reference [33] is related to cognitive impairment in euthymic bipolar disorders not schizophrenia.
Response 1. Thank you, it had to be a mistake. We have corrected that and now there is a reference that shows the neuroanatomical and neuropsychological deficits described in schizophrenia.
Point 2. In introduction part, authors could add some recent references (2020-2022)
Response 2. We agree with the suggestion, and we have updated some references in the introduction part.
Point 3. In 2.2.: add in the text Table 2 “Table 1 and table 2 show”.
Response 3. Thank you. It has been changed.
Point 4. In 2.4.: where are tables 1.2 and 2.1 mentioned in the text?
Response 4. Thank you for the question but I think there is a mistake here. There are not Table 1.2 and 2.1, there are Table 1 and 2 that show both search strategies and they are mentioned in 2.2. Search Strategy “Table 1 and Table 2 show the full search strategy used in the review.”
Point 5. In 2.4: “Two authors (X.X. and X.X.)”….. “third author (X.X.)” please add the name of the authors.
Response 5. Thank you for comment. We have added the name of the authors.
Point 6. In 2.5 : “two independent authors of the review (X.X. and X.X.)” please add the name of the authors.
Response 6. Thank you for comment. We have added the name of the authors.
Point 7. Please insert Tables 4, 4.1, 5, 5.1, 6 and 7 in the text.
Response 7. Regarding to the tables in the text, it was a problem with the format. We have done changes in the order and number, and I hope now it is solved.
Point 8. In 2.6: insert Table 3.1 next to “continued axis tool”
Response 8. Thank you for comment. We have added the name Table 3.1.
Point 9. Please put the tables 3, 3.1, 4, 4.1, 5, 5.1, 7 in other format, we don’t see them all!
Response 9. Regarding to the tables in the text, it was a problem with the format. We have done changes in the order and number, and I hope now it is solved.
Point 10. It would be better to insert the tables 3, 3.1, 4 and 4.1 in the results part
Response 10. About the Tables 3, 3.1., 4 and 4.1., Reviewer 2 suggested us to insert in Appendix part due to is a complementary information and is not related with the review method or the paper results. Therefore, we have decided to do it that way. We made two appendixes: Appendix A for the Appraisal tool for Cross-Sectional Studies (AXIS) about neurocognitive suicide and homicide markers in patients with schizophrenia spectrum disorders, and Appendix B for JBI checklist for longitudinal cohort and case-control studies to assess their methodological quality and to determine the extent to which a study addressed the possibility of bias in its design, conduct, and analysis. Now you can find those tables named as Table A1, A2, B1 and B2.
Point 11. In tables 3, 3.1, 4, 4.1, 5, 5.1, 6 and 7, it would be better to insert the reference number instead of the author's name.
Response 11. Thank you for the suggestion. We have added the reference number in each table.
Point 12. In 3.4.3. part, where is the Table 3.2 mentioned in the text?
Response 12. Thank you for the question. With the changes in the order and name of the tables, now the part 3.4.3. Shared neuropsychological impairments in suicidal and homicidal patients is shown in Table 4, where the readers could see graphically how two cognitive functions (processing speed and visual memory) could explain homicide and suicide behaviours.
Point 13. In discussion part, please add some recent references in studies related to fMRI
Response 13. We agree with the suggestion, and we have added some studies related to fMRI.
Point 14. Finally the most important point: there are several important limitations of the study (diagnosis, gender, sociodemographic and clinical characterics, methodologies of the studies, definition of suicidal and homicidal behaviors, neuropsychological assessments used etc.) and no strong conclusion can be established, so what is the interest of this study and the originality?
Response 14. Thank you for the comment, it is a very interesting point of view. I would like to express the interest of this study and the originality. First, it is a systematic review that shows and updates the information related with the neurocognitive markers in patients with schizophrenia spectrum disorders which have committed violent acts (last review was in 2014, Richard-Devantoy et al., 2014, and just insert studies till 2012. More than 10 years without updating this information, despite of there have been published several studies). Secondly, this study, unlike previous reviews, has analysed the quality of the papers included with AXIS and JBI tools. Third, the limitations showed the difficulties that exist in the neuropsychological study of patients with schizophrenia (different cognitive test and assess, pathologies, ages, clinical characteristics…), with the objective to guide future research. Finally, the main contribution of this review is that it tries to understand if there are shared cognitive mechanisms which can help to identify and predict risk areas in crime intervention and treatment programs.
Reviewer 2 Report
Thank you for giving me this opportunity to review this manuscript. the topic is interesting and methods is acceptable by and large, although the writing style should be improved substantially. My comments:
1. The title is not clear enough. It should be edited for more fluency and the type of review should be added.
2. There is a confusing discrimination between 'neurocognitive mechanism' and 'neurocognitive function' throughout the text. It seems that the objective of the study is neurocognitive functions of patients with schizophrenia and their relationship with suicidal or homicidal behaviors. It should be emphasized in every part of the text.
3. Definition of homicide should be revised based on a well-known scientific paper or reference and citation should be added.
4. The very last sentence of the introduction is not consistent with other reports from previous studies on suicide.
5. Novelty of the current systematic review should be mentioned substantially in comparison with previous ones.
6. Exact date of search should be added to the methods.
7. Tables 1&2 should be integrated.
8. Why DSM-IV? Almost all time of the review is after launching of DSM-5.
9. If the authors included patients with other psychotic disorders ie schizoaffective, schizophreniform, ..., 'Schizophrenia spectrum' should be used throughout the title and text.
10. Any abbreviation like AXIS should be used in full-term for first time.
11. '(n= 14) + 1 = N= 15' is not clear in the Fig-1. In addition, 'Records after duplicates removed (n= 391)' and 'Records screened(n= 391)' boxes should be integrated as one box. Same changed should be done in Fig-2.
12. Caption of tables should be informative enough. For instance, tables3&4.
13. I believe tables 3&4 should be revised and moved to appendix.
14. 'Tables 3 and 3.1' from the text. I can`t find 3.1.
15. ' 3.268' is not a good writing style and brings misunderstanding. '3268' is fine.
16. This sentence is not cleat at all: 'age was not specified in the remaining 12.5% (n=1) of the sample.'
17. What`s last paragraph of p. 17???
18. If the authors can conclude that which parts of cognitive functions are important in this group of patients as predictors, it should be clearly emphasized in the results, discussion and particularly conclusion.
Author Response
Thank you for giving us the opportunity to submit a revised draft of our manuscript. We appreciate the time and effort that you have dedicated to providing your valuable feedback on our manuscript. Here is a point-by-point response to the reviewers’ comments and concerns.
Point 1. The title is not clear enough. It should be edited for more fluency and the type of review should be added.
Response 1. We have changed the title:
Neurocognitive Suicide and Homicide Markers in Patients with Schizophrenia Spectrum Disorders: A Systematic Review
Point 2. There is a confusing discrimination between 'neurocognitive mechanism' and 'neurocognitive function' throughout the text. It seems that the objective of the study is neurocognitive functions of patients with schizophrenia and their relationship with suicidal or homicidal behaviors. It should be emphasized in every part of the text.
Response 2. Thank you for the comment, it is a very interesting point of view. The objective of the study is to identify the cognitive functions (memory, attention, executive functions, etc) of suicidal and homicidal behaviours in people with schizophrenia. When we talk about neuropsychological mechanisms, we refer to whether there is a set of cognitive functions or patterns which are shared in both behaviours.
We have changed the end of the introduction part; I hope it would be clear now.
Point 3. Definition of homicide should be revised based on a well-known scientific paper or reference and citation should be added.
Response 3. We have changed the definition based on a scientific paper of Paul R. Smit called Homicide data in Europe: Definitions, sources, and statistics. In Handbook of European homicide research: patterns, explanations, and country studies.
Point 4. The very last sentence of the introduction is not consistent with other reports from previous studies on suicide.
Response 4. As I told you in question 2, we have changed the end of the introduction. We didn’t want to be inconsistent with the previous information. We hope it would be clear now.
Point 5. Novelty of the current systematic review should be mentioned substantially in comparison with previous ones.
Response 5. Thank you for the comment. We have made changes in the end of the introduction part and in conclusions.
Point 6. Exact date of search should be added to the methods.
Response 6. We have added that information.
Point 7. Tables 1&2 should be integrated.
Response 7. Thank you for the suggestion but we don’t agree. There were two different search strategies: one for suicide or attempted suicide and another for homicide or violence. We consider that, despite of they are very similar, they must be in different boxes.
Point 8. Why DSM-IV? Almost all time of the review is after launching of DSM-5.
Response 8. Because the 100% of the included studies used DSM-IV to diagnosis schizophrenia spectrum disorders in their samples.
Point 9. If the authors included patients with other psychotic disorders ie schizoaffective, schizophreniform, ..., 'Schizophrenia spectrum' should be used throughout the title and text.
Response 9. Thank you for the comment. We have made those changes.
Point 10. Any abbreviation like AXIS should be used in full-term for first time.
Response 10. Now, it has been included into manuscript.
Point 11. '(n= 14) + 1 = N= 15' is not clear in the Fig-1. In addition, 'Records after duplicates removed (n= 391)' and 'Records screened(n= 391)' boxes should be integrated as one box. Same changed should be done in Fig-2.
Response 11. Thank you. We have added in 2.6. Synthesis of results the explanation about this. Regarding to the Figures, we have modified the boxes.
Point 12. Caption of tables should be informative enough. For instance, tables3&4.
Response 12. Now, Tables 3 and 4 are in the Appendix as you suggested in the comment 13.
We have renamed:
Appendix A1: Appraisal tool for Cross-Sectional Studies (AXIS)
Appendix A2: Continued AXIS tool.
Appendix B1: JBI checklist for longitudinal cohort studies.
Appendix B2: JBI checklist for longitudinal case-control studies.
Point 13. I believe tables 3&4 should be revised and moved to appendix.
Response 13. Thank you. We have modified this into manuscript.
Point 14. 'Tables 3 and 3.1' from the text. I can`t find 3.1.
Response 14. There was a mistake. Table 3.1. was “continued AXIS tool”. It is the continuation Appraisal tool for Cross-Sectional Studies (AXIS), now in Appendix A.
Point 15. ' 3.268' is not a good writing style and brings misunderstanding. '3268' is fine.
Response 15. Thank you, we have changed it.
Point 16. This sentence is not cleat at all: 'age was not specified in the remaining 12.5% (n=1)
of the sample.'
Response 16. We agree. Now we have modified it in the number of patients which the age has not been specified (n=398).
Point 17. What`s last paragraph of p. 17???
Response 17. There have been changes due to the format. Despite of that, I think you refer to the paragraph where we explain the neurobiology mechanisms which are related with a cognitive poor performance.
Point 18. If the authors can conclude that which parts of cognitive functions are important in this group of patients as predictors, it should be clearly emphasized in the results, discussion and particularly conclusion.
Response 18. Thank you for the suggestion. We have emphasized in the conclusions, writing again the cognitive functions that are important in this group of patients.
Round 2
Reviewer 1 Report
The authors have considered and responded to all my comments and suggestions.
Author Response
Thank you very much for the reviews provided and the interest shown.Reviewer 2 Report
Thank you for the resubmission. A few comments remaining:
1. '(n= 14) + 1 = N= 15' and '(n= 9) - 1 = N= 8' should be clear described as a legend of Fig1&2.
2. 'A systematic review of the recent literature was carried out to synthesize...' can bring misunderstanding. The sentence should be edited to show clearly, it`s about the current study.
3. I don`t know why number of some tables are with '.'. For instance, Table-3.1' should be Table-4 and number of others should be edited accordingly.
4. Ordering of studies in tables should be edited based on year or alphabetical ordering of names of first authors.
5. Year of study should be added to tables 5&5.1.
6. Number of references are not correct in the reference part.
7. Your response to point 8 of my previous comments should be added to the manuscript.
Author Response
Thank you for giving us the opportunity to submit a revised draft of our manuscript. We appreciate the time and effort that you have dedicated to providing your valuable feedback on our manuscript. Here is a point-by-point response to the reviewers’ comments and concerns.
Point 1. '(n= 14) + 1 = N= 15' and '(n= 9) - 1 = N= 8' should be clear described as a legend of Fig1&2.
Response 1. We agree and we have added that information as a legend in Fig1&2.
Point 2. 'A systematic review of the recent literature was carried out to synthesize...' can bring misunderstanding. The sentence should be edited to show clearly, it`s about the current study.
Response 2. Thank you. It has been changed.
Point 3. I don`t know why number of some tables are with '.'. For instance, Table-3.1' should be Table-4 and number of others should be edited accordingly.
Response 3. We agree with the suggestion. We have chanded that an now there aren’t tables with ‘.’.
Point 4. Ordering of studies in tables should be edited based on year or alphabetical ordering of names of first authors.
Response 4. Thank you for the comment. We have ordered the studies based on the year. From the most recent till the least recent.
Point 5. Year of study should be added to tables 5&5.1.
Response 5. We have added the years of the study in both tables, now named Table 6 and 7.
Point 6. Number of references are not correct in the reference part.
Response 6. After reviewing the reference’s part, all the numbers are correct. The changes that were made in the first revision were the following:
- We changed updating 3 references in the introduction (Shinko et al., 2020 [28]; Fjellvang et al., 2018 [35]; Schoretsanitis et al., 2019 [27]), in addition to changed reference [33] related with the homicide definition and reference [2] of World Health Organization (WHO) related with suicide prevalence.
- We added 3 references related to fMRI studies in discussion part (Potvin et al., 2018 [66]; Bohaterewicz et al., 2021 [65]; Tikàsz et al., 2018 [72]), reason why before there were 79 references and now there are 82.
I hope it would be clear now.
Point 7. Your response to point 8 of my previous comments should be added to the manuscript.
Response 7. Thank you for the suggestion. We have added that information to the manuscript. You can read it in part 3.4. Neurocognitive functioning in relation to homicidal or suicidal behaviour in schizophrenia.